# Type III Kounis Syndrome Secondary to Ciprofloxacin-Induced Hypersensitivity

**DOI:** 10.3390/medicina58070855

**Published:** 2022-06-26

**Authors:** Alberto Navarro-Navajas, Ingrid Casallas, Daniel Isaza, Paola Ortiz, Daniela Baracaldo-Santamaría, Carlos-Alberto Calderon-Ospina

**Affiliations:** 1Fundación Cardioinfantil, Bogotá 111221, Colombia; betonavarro87@gmail.com (A.N.-N.); ingridlizeth05@gmail.com (I.C.); disaza@lacardio.org (D.I.); 2Grupo Neuros, School of Medicine and Health Sciences, Universidad del Rosario, Bogotá 111221, Colombia; pao5336@hotmail.com; 3Pharmacology Unit, School of Medicine and Health Sciences, Universidad del Rosario, Bogotá 111221, Colombia; daniela.baracaldo@urosario.edu.co; 4GENIUROS Research Group, Center for Research in Genetics and Genomics (CIGGUR), School of Medicine and Health Sciences, Universidad del Rosario, Bogotá 111221, Colombia

**Keywords:** Kounis syndrome, hypersensitivity, acute coronary syndrome, ciprofloxacin, myocardial infarction

## Abstract

Kounis syndrome (KS) is a rare syndrome characterized by the co-occurrence of acute coronary syndromes in the setting of mast cell and platelet activation in response to hypersensitivity reactions. It can be manifested as coronary vasospasms, acute myocardial infarction, or stent thrombosis triggered by drugs, vaccines, foods, coronary stents, and insect bites. It is a life-threatening condition that needs to be adequately recognized for early diagnosis and appropriate treatment. In this case report, we present a 71-year-old patient with a history of arterial hypertension and non-ST elevation myocardial infarction six months earlier that was treated percutaneously with angioplasty plus stent implantation in the circumflex artery, who subsequently presented to the emergency department due to generalized itching associated with tongue swelling, dyspnea, and chest pain after ingestion of ciprofloxacin for the treatment of a urogenital infection. An electrocardiogram showed ST elevation in II, III, and aVF leads, and positive troponin; thus, a coronary arteriography was performed that showed complete thrombotic stent occlusion in the circumflex artery. Consequently, diagnosis of type 4b inferolateral acute myocardial infarction secondary to ciprofloxacin-triggered type III Kounis syndrome was made. The aim of this report is to understand the relationship between the allergic reaction to ciprofloxacin and the acute coronary syndrome, and to create awareness of the importance of early diagnosis and treatment of this potentially fatal syndrome.

## 1. Introduction

The association between hypersensitivity reactions and cardiovascular symptoms has been well documented, however it was not until 1991 when Kounis described that coronary arterial spasm or myocardial infarction could be provoked by an allergic reaction in response to excessive histamine release [1]. Kounis syndrome is now defined as a syndrome characterized by the co-occurrence of acute coronary syndromes in the setting of mast cell and platelet activation in response to hypersensitivity reactions. It can be classified into three subtypes: type I is coronary vasospasm without previous coronary abnormalities; type II is the presence of a vasospastic coronary event in the setting of an underlying atheromatous plaque that results in its erosion or rupture leading to coronary artery occlusion; type III occurs in the presence of a coronary artery stent, where the release of inflammatory mediators results in stent thrombosis [2]. It is a rare condition with an incidence of 0.0194% of all admissions and 3.4% of allergy patient admissions [3]. Although Kounis syndrome can occur at any age, more than half of the affected patients are 40–70 years old. Risk factors for Kounis syndrome include a history of previous allergy, hypertension, smoking, diabetes, and hyperlipidemia.

Many inflammatory mediators give rise to Kounis syndrome. Mast cells normally store ~500 secretory granules and have IgE attached to their surface. When these IgE antibodies encounter an allergen, they are crosslinked and cause the mast cell to degranulate. Among the inflammatory mediators contained in these granules are histamine, chemokines, tryptase, cathepsin-D, peptides, proteoglycans, arachidonic acid products, neutral proteases, and TNF-α. Histamine produces several manifestations in the cardiovascular system, mainly, coronary vasoconstriction, induction of tissue factor, platelet activation, modulation of the activity of neutrophils, monocytes and eosinophils, induction of proinflammatory cytokines, and intimal thickening [4,5]. On the other hand, neutral proteases are also secreted upon mast cell degranulation, and they can degrade the collagen cap, inducing erosion of the plaque and subsequent rupture [4]. Arachidonic acid products will further exacerbate the coronary vasoconstriction induced by histamine, and specifically thromboxane will also promote platelet aggregation [6]. Activated platelets in response to histamine will also contribute to a pro-thrombotic environment by releasing granules containing factor V, factor IX, and platelet activating inhibitor-1 [7]. All these mediators are released into the systemic circulation and can cause coronary artery spasm or thrombosis, leading to Kounis syndrome.

Kounis syndrome can be caused by a wide variety of insults. Drugs, substances, foods, environmental exposure, and certain conditions can trigger this syndrome. Many drug classes have been linked to this syndrome, including antibiotics, antivirals, antifungal, nonsteroidal anti-inflammatory drugs, proton pump inhibitors, antihypertensive drugs, and antihistamines, among others. However, the most frequently culprit drugs are penicillins [8] and cephalosporins [9,10]. Insect bites are the next most frequent cause of Kounis syndrome followed by antibiotics, bees, and wasps. Food is also a frequent cause, although not as common as antibiotics or insect bites. Shellfish, tuna fish, salt fish, uncooked anchovies, and kiwi appear to be the most frequently reported foods that can cause Kounis syndrome. There have also been reports of Kounis syndrome induced by intravenous contrast material [11], latex, and more recently, it has even been associated with COVID-19 disease and vaccines [12,13,14,15].

Even though antibiotic use is a frequent cause of Kounis syndrome that is mainly due to penicillin- or cephalosporin-induced allergy, Kounis syndrome secondary to the fluoroquinolone ciprofloxacin is not common. To the best of our knowledge, only three cases of ciprofloxacin-induced Kounis syndrome have been reported since 2009 [16,17]. This condition can be easily overlooked and may be the reason for the limited reports on ciprofloxacin as a trigger of Kounis syndrome. Following clinical manifestations (acute chest pain, palpitations and dyspnea, and allergic reactions such as erythematous rash, hives and wheezing), ECG, echocardiography, and coronary angiography are required to support the diagnosis of this disease [18].

The aim of this report is to create awareness of another potential trigger of Kounis syndrome not frequently encountered and increase clinical suspicion in clinicians to promote early diagnosis and appropriate treatment, which may affect the prognosis of the patient.

## 2. Case Presentation

This is a 71-year-old male with a history of arterial hypertension and non-ST elevation myocardial infarction six months earlier that was treated percutaneously with angioplasty plus stent implantation in the circumflex artery and medical therapy consistent in clopidogrel and aspirin. He was currently taking aspirin 100 mg PO once daily, atorvastatin 40 mg PO once daily, carvedilol 6.25 mg PO every 12 h, clopidogrel 75 mg PO once daily, omeprazole 20 mg PO once daily, and valsartan 160 mg once daily. He had no history of previous allergy to other penicillin-class antibiotics. He was admitted due to generalized itching associated with tongue swelling, dyspnea, and chest pain after the ingestion of ciprofloxacin for an acute urogenital infection 1–3 h before medical contact. He had never consumed ciprofloxacin before. The patient’s vital signs were: blood pressure, 90/50 mmHg; mean arterial pressure, 63 mmHg; heart rate, 100 beats per minute; respiratory rate, 27 breaths per minute; and oxygen saturation, 85%. No fever or other signs of sepsis were present. Eyelid edema, glossitis, and maculopapular rash on the thorax were found at the physical exam, along with hives. The cardiovascular exam was normal. The initial diagnosis was anaphylactic shock, and the treatment included 150 mg hydroxyzine, 200 mg hydrocortisone, subcutaneous adrenaline, and noradrenaline infusion. However, the patient developed retrosternal chest pain with oppressive characteristics, intensity 7/10. Electrocardiogram showed ST elevation in II, III, and aVF leads (Figure 1), and positive troponin.

With the diagnosis of acute coronary syndrome, a coronary arteriography was performed that showed complete thrombotic stent occlusion in the circumflex artery (Figure 2); therefore, the diagnosis of type 4b inferolateral AMI secondary to type III Kounis syndrome triggered by an allergic reaction to ciprofloxacin was made.

After making the diagnosis, primary percutaneous coronary intervention with angioplasty plus successful placement of a drug-eluting stent in the circumflex artery was made (Figure 2). An echocardiogram showed inferior hypokinesia with LVEF 45%. The medical therapy included antihistamine and systemic steroid for the allergic reaction, antithrombotic therapy, and advice to avoid the use of ciprofloxacin.

## 3. Discussion

Kounis syndrome is a difficult and underdiagnosed disease that should be considered in patients who have angina-like symptoms as well as systemic anaphylaxis. Clinical suspicion of Kounis syndrome is beneficial for early diagnosis and appropriate treatment, which may affect the prognosis. Among the three different variants of Kounis syndrome previously defined, our case corresponds to type III Kounis syndrome because the patient presented with pre-existing coronary artery disease and thrombosis in a drug-eluting coronary stent. Ciprofloxacin was considered the trigger of Kounis syndrome in this case because it has been previously documented [16,17], clinical symptoms appeared soon after intake, there are not many alternative causes that could account for the clinical, paraclinical and imaging findings, and thrombosis of the stent was evidenced by an objective medium. The adverse drug reaction probability scale (Naranjo scale) was designed to address causality of all types of adverse drug reactions [19]. It consists of 10 questions that can be answered as yes, no or not done/don’t know, for which different point values are assigned. In this way, a total of less than 2 points corresponds to low probability or doubtful, 2 to 4 is considered possible, 5 to 8 points is considered probable, and >9 is considered definite. The corresponding score for ciprofloxacin in our case was 6 points, so it can be classified as a probable cause.

As mentioned previously, Kounis syndrome is caused by inflammatory mediators released during allergic insults, and in type III Kounis syndrome, many of the compounds that form part of the stent can act as antigens. Coronary stents that are currently used include a bare metal stent, which has a stainless-steel platform that contains nickel, titanium, chromium, manganese, and molybdenum [20], and drug-eluting stents, which can contain cobalt or platinum, all of which can act as antigens and activate a strong immune response. However, the correlation between the time of ciprofloxacin intake and the occurrence of symptoms suggests ciprofloxacin induced an immune response that resulted in stent thrombosis in our case.

Differential diagnosis of Kounis syndrome includes Takotsubo myocardiopathy because it can produce hyperkinesia of the ventricle base and hypokinesia of the apex and midzone [21], which can also occur in Kounis syndrome as a result of inflammatory mediators. However, our patient’s echocardiogram showed inferior hypokinesia and Takotsubo myocardiopathy was discarded due to previous stent implantation in the circumflex artery and the accompanying signs and symptoms (eyelid edema, glossitis, generalized maculopapular rash, and itching). Furthermore, hypersensitivity myocarditis should be considered in the differential diagnosis. In both Kounis syndrome and hypersensitivity myocarditis, there is an allergic cause that affects the coronary arteries (in Kounis syndrome) or the heart muscle (hypersensitivity myocarditis) that can be difficult to differentiate clinically [22]. However, coronary artery involvement was demonstrated by coronary arteriography, clearly showing total stent occlusion in the distal circumflex artery accounting for the EKG abnormalities and the patient’s symptomatology. The remaining possible diagnosis was for an allergic reaction affecting the coronary arteries that could also cause stent thrombosis, thus the diagnosis of Kounis syndrome was made.

The diagnosis of Kounis syndrome is largely based on clinical signs and symptoms accompanied by laboratory findings that confirm the acute coronary insult (EKG, echocardiogram, angiography, troponin). A careful review of the patient’s history regarding previous allergies and current drug consumption should be made as 25% of patients that present with Kounis syndrome may have an allergy history. Laboratory exams that should be obtained in patients with suspected Kounis syndrome include a complete blood count (eosinophil abnormalities), cardiac enzymes and troponin I or T, C-reactive protein, IgE, histamine levels, arachidonic acid products, tumor necrosis factor, interferon, and interleukin 6, although normal levels of histamine or IgE do not exclude the diagnosis. Elevated troponin is seen in 60.6% of patients, ST elevation in inferior leads is seen in 67% of patients, and abnormal echocardiography can be seen in 58% of patients [23], all of which were present in our case.

Possible drug-drug interactions were also analyzed considering that the patient was also consuming aspirin, atorvastatin, carvedilol, clopidogrel, omeprazole, and valsartan [24]. Valsartan and ciprofloxacin have a class B risk rating since angiotensin II receptor blockers can enhance the arrhythmogenic effect of quinolones. Nonetheless, the symptomatology of the patient did not match this interaction. Ciprofloxacin and atorvastatin have no known drug interaction, nor carvedilol, clopidogrel, and omeprazole.

The treatment of Kounis syndrome is a challenge. The main goals are the treatment of anaphylactic shock and myocardial revascularization. Corticosteroids and H1 and H2 antihistamines are commonly used to decrease the allergic reaction [23]. In patients like ours with anaphylactic reactions, appropriate fluid resuscitation and epinephrine is also required for blood pressure support. In patients with types II and III Kounis syndrome, therapeutic management should also be based on acute coronary syndrome guidelines [25]. Aspiration of the intrastent thrombus with posterior histological analysis is recommended to visualize the presence of eosinophils (hematoxylin and eosin) and mast cells (Gemsa stain). Fortunately, patients with Kounis syndrome have a good prognosis, but an early diagnosis is necessary to provide better outcomes.

## 4. Conclusions

In conclusion, Kounis syndrome is a complex underdiagnosed and easily overlooked syndrome that requires immediate treatment. In some cases, it might be recommended to do an allergy workup to characterize food, drug, or insect bite allergies to prevent future events and to identify the culprit agent. Ciprofloxacin use should increase clinical suspicion of Kounis syndrome in the setting of angina-equivalent symptoms and systemic anaphylaxis. It is important to highlight that not every case of coronary vasospasm can be classified as Kounis syndrome. The individual should have subclinical, acute or chronic allergic reactions accompanied by cardiac symptomatology to be diagnosed with Kounis syndrome.

## Figures and Tables

**Figure 1 medicina-58-00855-f001:**
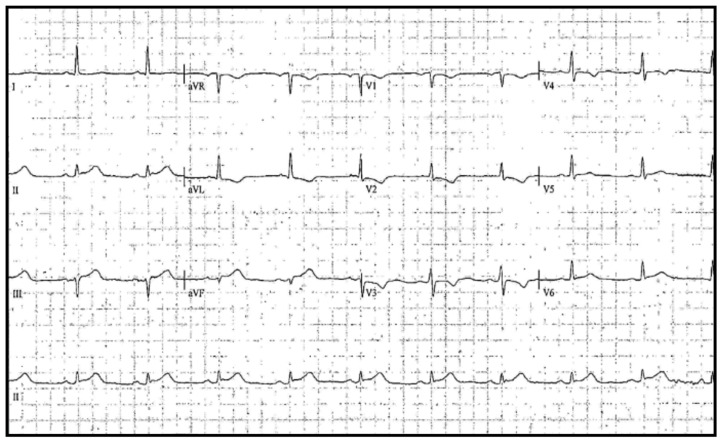
Electrocardiogram with sinus rhythm showing ST elevation in II, III, and aVF leads. Standard limb leads–I, II, III, augmented limb leads–aVR, aVL, and aVF, and precordial or chest leads–V1 to V6.

**Figure 2 medicina-58-00855-f002:**
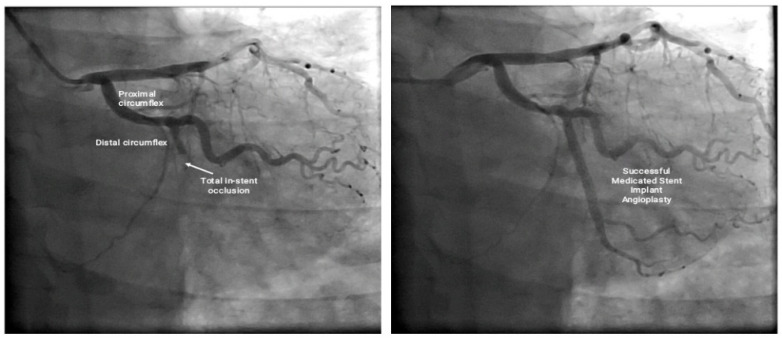
Coronary arteriography showing complete thrombotic stent occlusion in the circumflex artery (**left**), and successful posterior placement of a medicated stent in the circumflex artery (**right**).

## Data Availability

Not applicable.

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
