# Peer review of "Type III Kounis Syndrome Secondary to Ciprofloxacin-Induced Hypersensitivity"

_medicina, 2022, doi:10.3390/medicina58070855_

Round 1
Reviewer 1 Report
Please rephrased line 19 in abstract as Kounis syndrome can be manifested as ....not produce d.
Second clarify medicated stent ?
Do you mean drug eluting stent.
Overall good presentation of the case with minor English errors.
Author Response
Dear reviewer,
Thank you for taking the time to review our manuscript, we have made some changes and hope our manuscript is now ready for publication.
Please rephrased line 19 in abstract as Kounis syndrome can be manifested as ....not produced.
Answer: we have made the appropriate changes
Second clarify medicated stent ?
Do you mean drug eluting stent.
Answer: yes, we have made the correction
Reviewer 2 Report
More case details are necessary.
Specific comments:
1. "... with history of urogenital infection" - this can be omitted as it is an acute rather than chronic issue.
2. Given that ciprofloxacin is a very common antibiotic, has the patient taken it before and with no issues? Did the patient have issues with penicillin-class antibiotics, hence the choice in the first place?
3. Did the patient have a fever or appear septic at the time of presentation?
4. It is important to highlight that not every case of coronary vasospasm is Kounis syndrome and the individual should have subclinical/acute/chronic allergic reactions accompanied by cardiac symptomatology.
Author Response
Dear reviewer,
Thank you for taking the time to review our article, we have made some changes and hope our article is now ready for publication.
1."... with history of urogenital infection" - this can be omitted as it is an acute rather than chronic issue.
Answer: We have made the appropriate changes
- Given that ciprofloxacin is a very common antibiotic, has the patient taken it before and with no issues? Did the patient have issues with penicillin-class antibiotics, hence the choice in the first place?
Answer: the patient had never consumed ciprofloxacin before, no previous allergy to other penicillin-class antibiotics. We added this information to the case presentation.
- Did the patient have a fever or appear septic at the time of presentation?
Answer: the patient had no fever nor did he appear septic. We have included this information in the case presentation.
- It is important to highlight that not every case of coronary vasospasm is Kounis syndrome and the individual should have subclinical/acute/chronic allergic reactions accompanied by cardiac symptomatology.
Answer: we have included this information
Round 2
Reviewer 2 Report
It is important to state that the patient had never consumed ciprofloxacin before.
Author Response
Dear reviewer,
Thank you again for your comments, we have added that the patient had never consumed ciprofloxacin before in the case presentation.
Best regards.